# From Experts to a Generalist: Toward General Whole-Body Control for Humanoid Robots

**Yuxuan Wang**[1,2*]  **Ming Yang**[1,2*]  **Ziluo Ding**[2*]  **Yu Zhang**[2*]
**Weishuai Zeng**[1]  **Xinrun Xu**[2]  **Haobin Jiang**[1,2]  **Zongqing Lu**[1,2†]
[1]Peking University    [2]BeingBeyond
https://beingbeyond.github.io/BumbleBee/

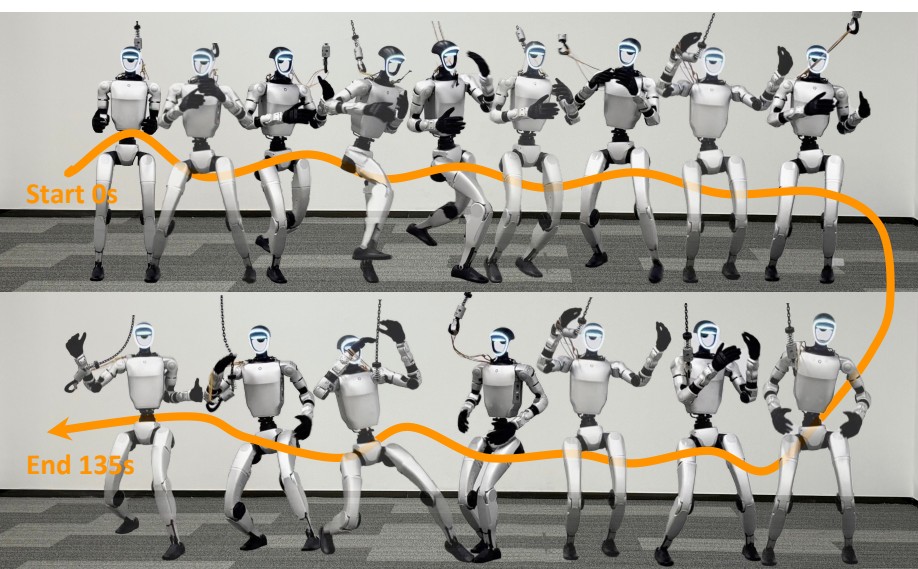

Figure 1: A general whole-body control policy in the real world tracks the motion of two consecutive long-duration actions, with a total duration of approximately 135 seconds. A boxing sequence appears at the top of the image, with a Charleston dance shown at the bottom.

## Abstract

Achieving general agile whole-body control on humanoid robots remains a major challenge due to diverse motion demands and data conflicts. While existing frameworks excel in training single motion-specific policies, they struggle to generalize across highly varied behaviors due to conflicting control requirements and mismatched data distributions. In this work, we propose BumbleBee (BB), an expert-generalist learning framework that combines motion clustering and sim-to-real adaptation to overcome these challenges. BB first leverages an autoencoder-based clustering method to group behaviorally similar motions using motion features and motion descriptions. Expert policies are then trained within each cluster and refined with real-world data through iterative delta action modeling to bridge the sim-to-real gap. Finally, these experts are distilled into a unified generalist controller that preserves agility and robustness across all motion types. Experiments on two simulations and a real humanoid robot demonstrate that BB achieves state-of-the-art general whole-body control, setting a new benchmark for agile, robust, and generalizable humanoid performance in the real world.

---

[*]These authors contributed equally to this work.

[†]Correspondence to Zongqing Lu <zongqing.lu@pku.edu.cn>.

39th Conference on Neural Information Processing Systems (NeurIPS 2025).

# 1    Introduction

Humanity has long dreamed of using humanoid robots to assist or replace humans in completing various daily tasks, including grabbing objects [Lenz et al., 2015, Joshi et al., 2020, Miller and Allen, 2004, Du et al., 2021, Shao et al., 2020], carrying goods [Pereira et al., 2002, Dao et al., 2024, Agravante et al., 2019], etc. At present, a single agile whole-body humanoid skill can be trained to a remarkably high level of performance. The ASAP [He et al., 2025] framework first trains motion-specific tracking policies in simulation, then refines them using real-world data through motion-specific dynamic policy learning. This framework enables the robot to effectively bridge the sim-to-real gap and achieve a wide repertoire of agile and expressive behaviors. However, **it is hard to directly apply this framework to general motion tracking**, as different motions require the robot to focus on different aspects of control. For example, aggressive actions like jumping or fast walking demand precise, high-torque control, while conservative motions prioritize balance and smoothness. These differences lead to mismatched data distributions across motion types. It can lead to conflicting gradients during training, ultimately degrading the performance of the policy.

Intuitively, to mitigate the issues caused by mismatched data distributions, there are two main directions. One is at the model level (*Be more powerful*): employing more expressive architectures such as Transformers or Diffusion Models [Shridhar et al., 2023, Kim et al., 2021, Brohan et al., 2022, Radosavovic et al., 2024], which are capable of capturing complex motion distributions across diverse skills. The other is at the data level (*Decompose the complexity*): introducing structure into the training data, such as clustering motions by type or difficulty, balancing data across motion categories, or using curriculum learning strategies to gradually expose the policy to increasingly diverse behaviors [Ji et al., 2024]. These approaches aim to reduce interference between conflicting samples and improve the generalization capability of the learned policy. Inspired by the success of mixture of experts (MoE) [Shazeer et al., 2017, Lepikhin et al., 2020] in LLMs, we follow the second path to build a general whole-body control policy.

To this end, we propose BumbleBee (BB), an **expert-generalist** pipeline that enables a general agile humanoid whole-body controller. Since both tracking and dynamic policies benefit from training on behaviorally similar motions, the dataset is first clustered using an autoencoder (AE) [Hinton and Salakhutdinov, 2006] trained on kinematic features, including additional leg-related features, and motion text descriptions. This design reflects the fact that different leg movements, such as walking, jumping, or crouching, require distinct torque control patterns, making leg dynamics a key factor in categorizing motion. A general tracking policy is initially trained on the full dataset and then fine-tuned within each cluster to obtain specialized **expert** policies. These experts are deployed on the real robot to collect real-world trajectories. For each cluster, a delta action model is trained to compensate for sim-to-real discrepancies by modeling the difference between simulated and real-world state transitions. The expert policies are further refined using the corresponding delta model in an iterative manner. Finally, a unified **generalist** is obtained through a combination of expert knowledge distillation.

To summarize, our main contributions are as follows: 1) We propose an expert-to-generalist framework for training agile whole-body control policies, which effectively mitigates interference across diverse motion types. 2) We introduce an auto-regressive clustering method that leverages leg-specific motion features and text descriptions to structure the dataset, enabling better specialization of control policies. 3) We demonstrate that our method, BumbleBee, achieves superior control performance in both simulation and real-world experiments, outperforming baselines in terms of agility, robustness, and generalization.

# 2    Related Work

## 2.1    Humanoid Whole-Body Control

Whole-body control (WBC) [Chignoli et al., 2021, Darvish et al., 2019, Dallard et al., 2023, Hutter et al., 2016, Kajita et al., 2001, Moro and Sentis, 2019, Li et al., 2025, Ding et al., 2025] is essential for high DoF humanoid robots to perform coordinated and stable movements in complex tasks. Recent advances in reinforcement learning have enabled robots to acquire diverse whole-body skills, yet training a general WBC policy remains challenging due to hardware constraints and the inherent complexity of the robotic action space.

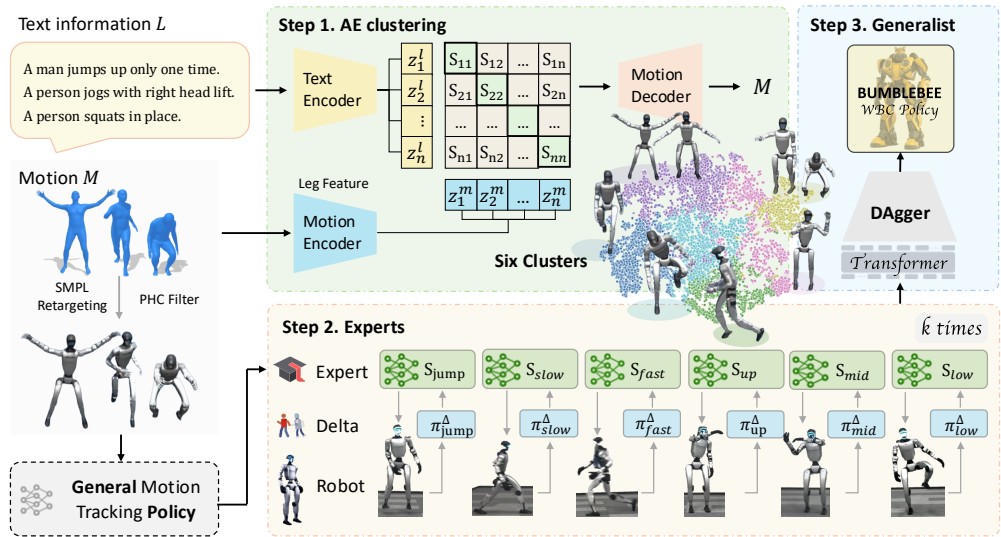

Figure 2: Overview of the BumbleBee framework. The left section illustrates the data curation stage, which includes motion retargeting and PHC-based filtering. The upper middle part shows the autoencoder-based clustering process, where motions are grouped via their semantic and kinematic characteristics. The lower section depicts the iterative delta fine-tuning of expert policies for each cluster. Finally, as shown on the far right, all experts are distilled into a single general WBC policy using a Transformer-based architecture.

Previous works address this in different ways: the H2O series [He et al., 2024a,b] incorporates global keypoint positions but suffers from error accumulation in long-horizon tasks; HumanPlus [Fu et al., 2024] uses only joint angles for better real-world adaptability; Hover [He et al., 2024c] introduces observation masking for flexible control mode switching. However, these methods are all trained on the full AMASS [Mahmood et al., 2019] dataset, where diverse motion distributions may cause gradient conflicts and reduce generalization. Exbody2 [Ji et al., 2024] attempts to mitigate this through progressive learning based on task difficulty.

In contrast, we propose a clustered expert-to-generalist paradigm. By classifying motion types and training specialized expert policies on each cluster before distillation, our approach reduces cross-task interference and improves both training efficiency and generalization.

## 2.2  Sim-to-Real Transfer

The sim-to-real transfer challenge primarily arises from two factors: (1) inaccuracies in robot modeling, including structural simplifications and parameter errors, and (2) the complex, nonlinear, and time-varying dynamics of real-world environments, *e.g.,* contact events, friction variations, and elastic deformations, that exceed the fidelity of modern physics simulators.

Traditional approaches address this gap through System Identification (SysID) [Åström and Eykhoff, 1971, Kozin and Natke, 1986, Yang et al., 2024, Yu et al., 2018, 2020], which uses real-world observations to calibrate simulator parameters for improved physical realism. Domain Randomization offers an alternative by perturbing environment parameters during training to enhance policy robustness and generalization.

ASAP proposes a different strategy by learning an action delta model to directly compensate for simulation-reality discrepancies using real-world data. Building on this idea, our method extends the action delta model by training it separately for each motion cluster. Compared to a single unified model, this cluster-specific approach enables improved sim-to-real performance across a diverse range of motions.

# 3 Method

Training a general whole-body control policy directly is challenging, primarily due to conflicts among diverse motion types. These motions differ significantly in their control objectives, for example, jumping requires high-torque control, whereas in-place movement emphasizes balance and continuity. To address this challenge, we take a new perspective on organizing motion data and policy learning by introducing an expert-to-generalist framework as illustrated in Figure 2. Rather than training a single policy on a heterogeneous dataset, we first partition the data based on the kinematic characteristics of each motion type, thereby reducing cross-task interference and enabling more targeted learning.

**Dataset Curation.** Following prior work [He et al., 2024a], we first retarget SMPL-format [Loper et al., 2015] human motion sequences from the AMASS dataset into robot-specific representations, including global translations and joint axis-angle rotations. Given that AMASS contains a wide range of motions, such as crawling and climbing, we perform an additional data cleaning step to ensure quality. We apply PHC [Luo et al., 2023] to filter the dataset [He et al., 2025], resulting in 8,179 high-quality trajectories for clustering and training.

## 3.1 AE Clustering

We leverage intermediate representations generated within a self-supervised framework, *i.e.* an autoencoder, to cluster motion data from the entire dataset. We aim to categorize the data based on motion types, for example, grouping jumping movements into one cluster and in-place movements into another.

Our method not only relies solely on autoencoders to reconstruct full motion sequences, but also incorporates textual annotations to align motions at the semantic level. This ensures that motions with diverse patterns but shared semantics are positioned closer in the latent space. For example, the act of walking may take the form of linear movement or circular paths. These patterns would be hard to associate with based only on motion alignment. However, things can be different when textual semantic alignment is introduced. Note that the text information for training is from the open-source HumanML3D [Guo et al., 2022] dataset, which offers textual annotations and corresponding frame ranges for most of the sequences in AMASS.

In addition, our method does not reconstruct motion sequences represented by the SMPL format. Since SMPL primarily contains joint angles and root transformations, it fails to represent the kinematic dynamics essential for distinguishing motion types. To address this, we first apply forward kinematics to convert joint rotations and root positions into 3D coordinates of joints in the world frame. Subsequently, we prune redundant joints and introduce foot velocity relative to the world frame, enhancing the model's capacity to differentiate between motion types such as jumping, standing, and walking.

For the encoding process, we adopt a Transformer for motion encoding, following the previous work [Petrovich et al., 2023, 2022]. The motion encoder takes as input a motion sequence $M_{\text{full}} = \{\mathbf{p}_t, \mathbf{r}_t, \dot{\mathbf{r}}_t, \mathbf{c}_t, \mathbf{v}_t^{\text{feet}}\}_{t=1}^{T}$ and outputs a motion representation $z^m$, where $\mathbf{p}_t \in \mathbb{R}^{N \times 3}$ represents the 3D positions of all joints, $\mathbf{r}_t \in \mathbb{R}^3$ is the root translation in 3D space, $\dot{\mathbf{r}}_t \in \mathbb{R}^3$ is the root velocity, $\mathbf{c}_t \in \{0,1\}^F$ denotes the binary contact states of $F$ feet, and $\mathbf{v}_t^{\text{feet}} \in \mathbb{R}^{F \times 3}$ represents the 3D velocities of the $F$ feet. Textual data $l$ is first processed through a BERT model [Devlin et al., 2019] for serialization and then passed through a Transformer to obtain a latent representation $z^l$ with the same dimensionality as $z^m$.

The decoding module uses the same Transformer as the motion encoder, but with different input and output dimensions. The reconstruction focuses only on a subset of key joints (*i.e.* head, pelvis, hands, and feet), allowing the model to concentrate on the core motion features. Specifically, it reconstructs the same features of the motion inputs from the latent representation $z^l$ or $z^m$, except that some redundant joint 3D positions mentioned above are removed. The loss function is defined as:

$$\mathcal{L}_{\text{cluster}} = \mathcal{L}_{\text{InfoNCE}}(z^l, z^m) + \mathcal{L}_2(z^l, z^m) + \mathcal{L}_{\text{huber}}(\hat{M}^l, M) + \mathcal{L}_{\text{huber}}(\hat{M}^m, M),$$

where $z^l$ and $z^m$ denote the intermediate latent variables for the text and motion modalities, respectively. $\mathcal{L}_{\text{huber}}$ refers to the Huber loss function, $M$ represents key features selected from the

ground-truth motion sequence $M_{\text{full}}$. $\hat{M}^l$ and $\hat{M}^m$ are the reconstructed features from the text and motion modalities, respectively. The first two terms are used to align the motion latent space with the semantic latent space, and the last two terms are reconstruction losses to train the autoencoder. Finally, we apply the K-means algorithm to cluster the latent variables of all motion data, generated by the learned motion encoder.

## 3.2 Experts

To improve motion specialization and sim-to-real transfer, we introduce expert policies, *i.e.* motion tracking and delta action policies, trained on motion clusters derived from AE embeddings. All models in this part are implemented using a three-layer multilayer perceptron (MLP) and are trained via reinforcement learning (RL). More details of reward design can be found in the Appendix B.

### 3.2.1 Motion tracking training

Initially, a general motion tracking policy is trained on the entire dataset as a base model. Note that both joint angles and selected keypoint positions of the tracking/reference motion are included in the policy's observation. More details about the observation design of our framework are included in the Appendix A.

Instead of training from scratch, all expert motion tracking policies are derived from the base model. This is because we expect each expert to retain some generalization capability for other motion clusters. Each expert policy is then fine-tuned on a specific motion cluster, allowing the policy to specialize in a behaviorally consistent cluster of skills, *e.g.,* walking, standing, or jumping. This fine-tuning significantly improves motion fidelity and effectively resolves training conflicts caused by diverse motion types.

### 3.2.2 Multi-stage delta action training

To further overcome the sim2real gap for tracking policy, we adopt the delta action fine-tuning framework. By applying the delta action on the simulator dynamics and continuing to fine-tune the tracking policy within this modified environment, the training process effectively approximates the training directly in the real world. *Our key contribution lies in specializing this framework by training expert delta models for each motion cluster, rather than relying on a single unified model.*

In more detail, each expert motion tracking policy is deployed on the real robot to collect its corresponding motion trajectories in the real-world environment. At each timestep $t$, using the onboard sensors of the 29-DoF Unitree G1 robot (23 DoF actively controlled, excluding the wrist joints), we record the following: base linear velocity $v_t^{\text{base}} \in \mathbb{R}^3$, the robot base orientation as a quaternion $\alpha_t^{\text{base}} \in \mathbb{R}^4$, the base angular velocity $\omega_t^{\text{base}} \in \mathbb{R}^3$, the vector of joint positions $q_t \in \mathbb{R}^{23}$, and joint velocities $\dot{q}_t \in \mathbb{R}^{23}$. For each expert, we randomly sample several dozen reference motions for repeated execution, resulting in the collection of over a hundred real-world trajectories in total. Subsequently, we further train expert delta action models based on the data collected for each expert following ASAP [He et al., 2025]. We find that, due to the consistent dynamic characteristics within each cluster of motion, this expert-specific training significantly improves the fitting accuracy of the delta action models and enables more efficient correction of the sim-to-real gap. Compared to a general delta action model, the expert models exhibit notable advantages in motion compensation accuracy and overall control performance.

With the learned delta action models, $\pi^\Delta(s_t, a_t)$, we reconstruct the simulation environment as follows, $s_{t+1} = f^{\text{sim}}(s_t, a_t + \pi^\Delta(s_t, a_t))$ and fine-tune the pretrained expert motion tracking policies within this modified environment. This process can be performed iteratively until both kinds of expert policies converge.

## 3.3 Generalist

After optimizing the expert policies, we employ knowledge distillation to integrate the knowledge of each expert policy and generate a general whole-body control policy. We adopt DAgger [Ross et al., 2011] to implement multi-experts distillation. The distillation loss function is defined as:

$$\mathcal{L}_{\text{distill}} = \mathbb{E}_{s \sim \mathcal{D}} \left[ \text{KL} \left( p_{\text{general}}(a \mid s) \, \| \, p_{\text{expert}, k(s)}(a \mid s) \right) \right]$$

Table 1: Main results evaluated in IsaacGym and MuJoCo. We assess performance using three key metrics: Success Rate (SR), Mean Per Joint Position Error (MPJPE), and Mean Per Keypoint Position Error (MPKPE). BB demonstrates superiority over baselines.

| Method | IsaacGym | | | MuJoCo | | |
|---|---|---|---|---|---|---|
| | SR↑ | MPKPE↓ | MPJPE↓ | SR↑ | MPKPE↓ | MPJPE↓ |
| OmniH2O [He et al., 2024a] | 85.65% | 87.83 | 0.2630 | 15.64% | 360.96 | 0.4601 |
| Exbody2 [Ji et al., 2024] | 86.63% | 86.66 | 0.2937 | 50.19% | 272.42 | 0.3576 |
| Hover [He et al., 2024c] | 63.21% | 105.84 | 0.2792 | 16.12% | 323.08 | 0.3428 |
| General Init | 88.69% | 92.88 | 0.2628 | 33.01% | 342.68 | 0.3467 |
| **BumbleBee (BB)** | 89.58% | 83.30 | 0.1907 | 66.84% | 294.27 | 0.2356 |

where $\mathbb{E}_{s \sim \mathcal{D}}$ denotes the expectation over the states $s$ in the training dataset $\mathcal{D}$, $\mathrm{KL}(\cdot \| \cdot)$ is the Kullback-Leibler divergence, $p_{\mathrm{expert}, k(s)}$ is the expert policy corresponding to state $s$, and $p_{\mathrm{general}}$ is the general policy. Here, $k(s)$ denotes the index of the expert that corresponds to the cluster to which state $s$ belongs.

However, we observed that the capacity of the three-layer MLP is limited, making it insufficient to effectively learn behaviors of multiple expert policies. To address this issue, we adopt a more expressive architecture, the Transformer, to serve as the backbone for the final general policy model. The Transformer enables better modeling of complex patterns across diverse states, allowing for more effective fusion of expert knowledge. More details of the architecture can be found in the Appendix C.

# 4 Experiment

## 4.1 Experiment Setup

Our models are trained in IsaacGym. Since there is a large gap between IsaacGym and the real world, MuJoCo serves as a more reliable proxy for evaluating the capability of the model. As a result, most of our evaluations are conducted in MuJoCo. We assess performance using the filtered AMASS dataset described in Section 3. For real-world testing, we further include long-range motions to evaluate the generalization and tracking capabilities of BumbleBee (BB). All training and deployment are performed on the Unitree G1 robot, which has 29 DoF, 23 of which are actively controlled, excluding the wrist joints.

**Baselines.** To evaluate the effectiveness of our method, we compare it with three state-of-the-art (SOTA) methods: OmniH2O [He et al., 2024a], Exbody2 [Ji et al., 2024], and Hover [He et al., 2024c]. For the fair comparison, we either use the officially released code or closely follow the official implementation when code is unavailable, adapting each method to the Unitree G1 robot (instead of H1/H1-2). In addition, we keep the training dataset consistent with that used for BB. For Hover, we use unmasked observations during the evaluation to ensure optimal performance.

**Metrics.** We assess performance using three key metrics: Success Rate (SR), Mean Per Joint Position Error (MPJPE), and Mean Per Keypoint Position Error (MPKPE). SR reflects the overall capability of the policy. MPJPE measures its accuracy in tracking retargeted joint angles, while MPKPE evaluates the precision of tracking retargeted keypoints in the world coordinate frame. Among all metrics, *SR is the most critical, as it reflects the overall viability and stability of the policy.* Other metrics, such as MPJPE and MPKPE, are only meaningful when the policy can successfully complete the task, as indicated by a high SR. Detailed definitions of these metrics can be found in Appendix A.

## 4.2 Cluster Analysis

*How do we determine the number of clusters?* We determine the number of clusters using the Elbow Method [Thorndike, 1953], which identifies the trade-off point between using fewer clusters and

Table 2: Statistics for each cluster. We measured the motion sequences in terms of the displacement distance (Displacement), the movement speed (Speed), the average displacement along the z-axis (Z-Move), the average per-step axis-angle change of all joints (Joint), as well as the joint variations of the upper and lower body (Lower and Upper Joint).

| Cluster | Disp (m) | Z-Move (mm) | Speed (m/s) | Joint | Lower Joint | Upper Joint | Keywords |
|---------|----------|-------------|-------------|-------|-------------|-------------|----------|
| *Jump* | 2.32 | 9.21 | 0.329 | 0.379 | 0.346 | 0.405 | jumps, jumping |
| *Walk-slow* | 3.42 | 1.60 | 0.353 | 0.279 | 0.358 | 0.215 | jogs, runs |
| *Walk-fast* | 3.24 | 5.63 | 0.429 | 0.430 | 0.472 | 0.396 | walks, forward |
| *Stand-up* | 0.89 | 0.68 | 0.061 | 0.230 | 0.136 | 0.305 | something, hand |
| *Stand-mid* | 1.33 | 0.82 | 0.119 | 0.198 | 0.170 | 0.221 | arms, hand |
| *Stand-low* | 1.84 | 1.52 | 0.148 | 0.274 | 0.281 | 0.267 | foot, leg |

minimizing the within-cluster sum of squares (total square distance from all points to their respective cluster centers). Based on this analysis, we set the number of clusters to six, as shown in Figure 3.

***Whether the clustering results are meaningful in terms of both kinematics and semantics?*** Table 2 summarizes the kinematic features of the six clusters. The first cluster has the highest Z-Move value (average displacement along the z-axis), and is therefore categorized as the jump cluster. The second and third clusters exhibit the largest ranges of displacement and are classified as walking behaviors. However, the third cluster demonstrates higher speed compared to the second. The remaining three clusters primarily consist of in-place movements, based on their lower speeds and smaller displacement ranges. More specifically, the fourth cluster involves larger upper-body movements, the sixth is characterized by more prominent lower-body movements, and the fifth falls between the two. From the semantic perspective, we extracted the top keywords from each cluster and find that their meanings closely align with the corresponding kinematic characteristics, demonstrating that the clusters are semantically meaningful. For convenience, all clusters are named exactly as specified in Table 2.

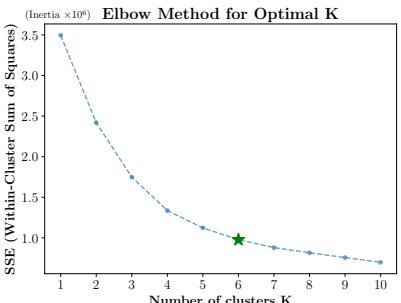

Figure 3: Elbow method showing the trade-off between cluster number and within-cluster sum of squares, with $K = 6$ selected as our selected point.

## 4.3 Generalist Performance

***Does our expert-to-generalist framework demonstrate superiority over existing methods?*** We analyze the performance of BB compared to existing SOTA methods in both IsaacGym and MuJoCo simulators. As shown in Table 1, BB outperforms all baselines across nearly all evaluation metrics. In IsaacGym, BB shows advantages over other methods in terms of success rate, MPJPE, and MPKPE; however, the minor differences observed are within reasonable variation and likely attributable to differences in hyperparameter tuning rather than the algorithm itself. In contrast, the performance gap becomes more pronounced in MuJoCo, a more realistic simulator that better reflects real-world dynamics. In this setting, BB achieves a success rate of 66.84%, significantly outperforming Exbody2 (50.19%) and all other baselines, which fall below 40%. This highlights the strong generalization capability of BB, which effectively integrates the specialized strengths of multiple experts into a unified policy, enabling stable and accurate execution of whole-body motions across different domains.

Table 3: Comparison of BB with a general policy trained without experts and one trained on experts from randomly split clusters, evaluated by success rate.

|  | IsaacGym | MuJoCo |
|--|----------|--------|
| *General Init* | 88.69% | 33.01% |
| *Random* | 86.25% | 35.36% |
| **BB** | 89.58% | 66.84% |

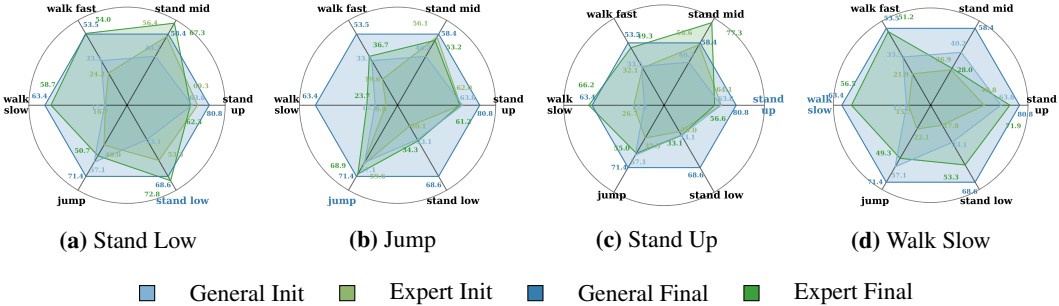

**(a)** Stand Low  **(b)** Jump  **(c)** Stand Up  **(d)** Walk Slow

□ General Init  □ Expert Init  □ General Final  □ Expert Final

Figure 4: Evaluation of expert vs. generalist models in MuJoCo, measured by success rate.

**Whether the clustering is beneficial for training?** To explore this question, we conduct two ablation studies. The first baseline, *General Init*, directly trains a generalist policy without any expert specialization. The second, *Random*, trains a generalist policy based on experts derived from six randomly partitioned subsets of the data. As shown in Table 3, the *Random* offers no clear advantage over *General Init*, highlighting the limited benefit of naive data partitioning. The superior performance of BB demonstrates that reasonable clustering plays a crucial role in policy learning. Even within randomly divided subsets, conflicting motion patterns may impede effective RL exploration.

## 4.4 Expert Performance

**Do expert policies outperform the generalist within their respective motion clusters?** To answer this question, we evaluate the performance of expert policies. In Figure 4, we show performance improvements across four motion clusters during multi-stage delta action training, with results for the remaining clusters included in Appendix D. Specifically, we compare success rates in MuJoCo across four variants: the generalist policy without expert specialization (*General Init*), expert policies before delta action fine-tuning (*Expert Init*), expert policies after two fine-tuning rounds (*Expert Final*), and the final distilled generalist (*General Final*).

From these results, we make several key observations. First, training expert policies on motion clusters leads to stronger task-specific performance compared to the *General Init*, validating the effectiveness of our autoencoder-based clustering and the importance of specialization. Second, experts maintain a degree of generalization to motions outside their clusters, likely due to being initialized from the *General Init*. Third, delta action fine-tuning significantly improves each expert's performance. Surprisingly, the final generalist (*General Final*), distilled from all experts, sometimes outperforms the individual experts, *i.e. jump* and *walk-slow*. These motions are initially difficult to execute stably. We hypothesize that the generalist benefits from inheriting stable control behaviors from multiple experts, allowing it to execute challenging motions more reliably.

**How does iterative delta fine-tuning affect the motion tracking policy?** We adopt an iterative approach to the optimization process, based on the intuition that each refinement improves the tracking policy, which enables the collection of higher-quality real-world data, thereby allowing the delta action policy to be further enhanced and, in turn, further improve the tracking policy. To validate this, we evaluate policy performance across three training stages in MuJoCo and also verify the results on a real humanoid robot. As shown in Table 4, the mean success rate steadily increases from $51.49\%$ (*Iter 0*) to $60.33\%$ (*Iter 1*), and further to $70.37\%$ (*Iter 2*), clearly demonstrating the effectiveness of iterative refinement. While additional iterations could potentially yield even better results, our experiments are constrained by limited computational resources.

We take *stand-low* as an example to provide a more intuitive understanding. As shown in Figure 5, without delta action fine-tuning (*Iter 0*), the policy fails to stabilize the foot, resulting in landing failure and deployment breakdown. After the first iteration (*Iter 1*), landing stability improves significantly, although the robot still struggles to lift its feet and exhibits visible trembling. After the second iteration (*Iter 2*), the policy can track the motion smoothly and maintain balance. More detailed visualizations can be found in the Appendix D.

**Whether the delta action model needs to be trained in a specialized manner?** To address this question, we conduct an ablation study using all collected real-world data to train a general delta action model. The results are summarized in Table 5, where *Expert Gen Final* represents the final

Table 4: Mean success rates of experts evaluated on their respective motion clusters. The evaluation is in MuJoCo.

|  | Iter 0 | Iter 1 | Iter 2 |
|---|---|---|---|
| SR | 51.49% | 60.33% | 70.37% |

Table 5: Ablation Results of the General Delta Action Model on three clusters. *Expert Gen Final* represents the final expert after fine-tuning on the general delta model.

| Cluster | Expert Init | Expert Gen Final | Expert Final |
|---|---|---|---|
| *Jump* | 59.64% | 50.71% | 68.92% |
| *Stand-up* | 64.11% | 75.85% | 77.32% |
| *Walk-slow* | 15.71% | 42.32% | 56.50% |

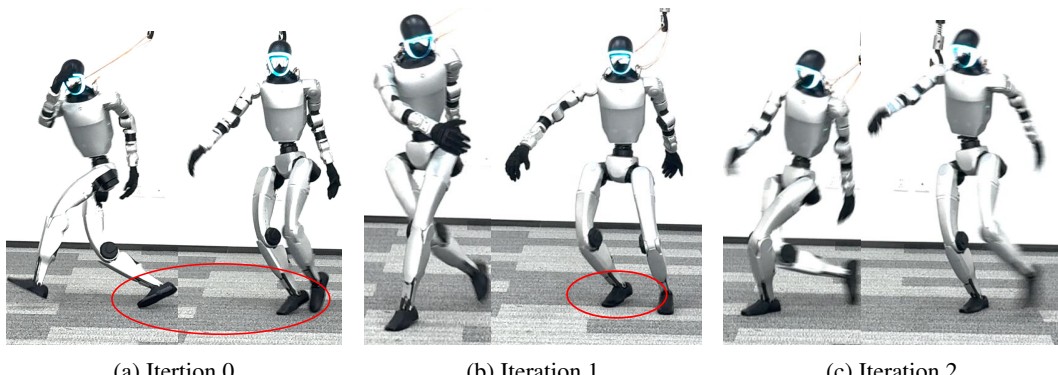

| (a) Itertion 0 | (b) Iteration 1 | (c) Iteration 2 |
|---|---|---|

Figure 5: Visualization of expert performance across iterations in the real world. We deploy the *stand-low* policies from three training iterations to perform the same right-stepping motion on a real robot, and observe a clear improvement in foot stability with each iteration.

expert after fine-tuning on the general delta model. The general delta action model is trained using all real-world data collected in each iteration loop.

We observe that the general delta model improves performance in two motion clusters but struggles significantly on *Jump*. This aligns with our hypothesis that delta action learning is sensitive to distributional shifts across motion types, making the single general delta action model less effective in highly diverse categories. Furthermore, we find that the general model is harder to train and less stable compared to its specialized counterparts.

***Why does a single delta action model struggle to generalize across motion clusters?*** To further investigate the variation in delta actions across different motion clusters, we analyze the magnitude of the delta action outputs. As shown in Table 6, *Stand-up* exhibits the smallest delta values, which is expected given that its motions are largely static. In contrast, *Walk-fast* shows the highest delta magnitudes due to its inherently large motion amplitudes. These distributional shifts across clusters introduce conflicts when attempting to train a single delta action model, further highlighting the need for cluster-specific modeling.

Table 6: Magnitudes of delta action outputs for the ankle joint across different clusters. Higher values indicate greater motion adjustments.

|  | *Jump* | *Walk-slow* | *Walk-fast* |
|---|---|---|---|
| Yaw | 0.2107 | 0.2865 | 0.3399 |
| Roll | 0.0589 | 0.0956 | 0.0843 |
|  | *Stand-up* | *Stand-mid* | *Stand-low* |
| Yaw | 0.1556 | 0.2098 | 0.2599 |
| Roll | 0.0278 | 0.0509 | 0.1273 |

## 5 Conclusion

We present BB, a framework for training agile and general whole-body control policies on humanoid robots. It follows an expert-to-generalist framework and uses auto-regressive encoder to cluster motions by semantics and leg dynamics, enabling effective specialization and knowledge transfer. Extensive experiments show that BB outperforms baselines in agility, robustness, and generalization, highlighting its potential for real-world deployment.

**Limitation** Currently, BB does not utilize additional high-precision localization sensors such as GPS or Visual-Inertial Odometry (VIO) [Huai and Huang, 2022]. As a result, it lacks access to global positioning information, which may introduce biases when aligning with the reference motion sequence. We believe that equipping BB with additional sensors like a high-precision IMU could significantly improve its performance in real-world scenarios by providing more accurate and reliable pose estimation. Moreover, the complexity of the overall pipeline in BB constrains its scalability, particularly when integrating real-world training feedback.

## Acknowledgments

This work was supported by NSFC in part under Grant 62450001 and 62476008. The authors would like to thank the anonymous reviewers for their valuable comments and advice.

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

# A  Environment Details

## A.1  RL environment

We provide a detailed training and test environment setting in this subsection.

**Observation**  For Privileged observation, we use proprioception, including linear velocity, angular velocity, joint position, joint velocity, and last action, and task-relevant observation, including target joint positions, target keypoint positions, target root translations, and target root rotations in the global coordinates. For student policies, we use all proprioception observation, except for linear velocity. For task-relevant information, we only preserve target joint positions, root translation, and root rotations in the local coordinates. For teacher policy, we take observations from 5 timesteps as input, and for student policies, we take observations from 10 timesteps as input.

For the delta action policy, we use the full proprioception of the teacher policy mentioned above, as well as the tracking policy actions. Note that we don't use the global information like root position and keypoint positions.

**Action**  We use roportional derivative (PD) controller to control the 23 DoF of the G1 (totally 29 DoF). And the policy outputs are the target joint position for PD controller.

**Termination**  In addition to falls, we added an additional termination condition during the training and testing process, where the position of the keypoints must not exceed a threshold. During training, the threshold is set from 0.8 down to 0.3 using curriculum learning. During testing, a threshold of 0.8 is used for walking, and 0.4 is used for the other tasks.

Table 7: Reward design for tracking policy.

| Term | Expression | Weight |
|---|:---:|---:|
| **Penalty** | | |
| Torque limits | $\mathbf{1}(\tau_t \notin [\tau_{\min}, \tau_{\max}])$ | $-10$ |
| DoF position limits | $\mathbf{1}(d_t \notin [q_{\min}, q_{\max}])$ | $-10$ |
| DoF velocity limits | $\mathbf{1}(\dot{d}_t \notin [\dot{q}_{\min}, \dot{q}_{\max}])$ | $-10$ |
| **Regularization** | | |
| DoF acceleration | $\|\ddot{d}_t\|_2^2$ | $-3 \times 10^{-8}$ |
| Action rate | $\|a_t - a_{t-1}\|_2^2$ | $-2$ |
| Action smoothness | $\|\dot{a}_t - \dot{a}_{t-1}\|_2^2$ | $-2$ |
| Torque | $\|\tau_t\|$ | $-0.0001$ |
| Stumble | $\mathbf{1}(F_{\text{feet}}^{xy} > 5 \times F_{\text{feet}}^z)$ | $-0.00125$ |
| Feet orientation | $\sum_{\text{feet}} \|\text{gravity}_{xy}\|$ | $-2.0$ |
| **Task reward** | | |
| Body position | $\exp\left(-4 \cdot \|\hat{p}_{\text{body}} - p_{\text{ref}}\|\right)$ | $1.0$ |
| Root rotation | $\exp\left(-4 \cdot \|q_{\text{toot}} - q_{\text{root}}\|\right)$ | $0.5$ |
| Root angular velocity | $\exp\left(-4 \cdot \|\omega_{\text{body}} - \omega_{\text{ref}}\|\right)$ | $0.5$ |
| Root velocity | $\exp\left(-4 \cdot \|v_{\text{root}} - v_{\text{ref}}\|\right)$ | $0.5$ |
| DoF position | $\exp\left(-4 \cdot \|d - d_{\text{ref}}\|\right)$ | $0.5$ |
| DoF velocity | $\exp\left(-4 \cdot \|\dot{d} - \dot{d}_{\text{ref}}\|\right)$ | $0.5$ |

## A.2  Deployment

The policy runs at an inference frequency of 50 Hz. The low-level interface operates at 200 Hz, ensuring smooth real-time control. Communication between the control policy and the low-level interface is facilitated via Lightweight Communications and Marshalling (LCM).

### A.3 Metrics

**Success Rate (SR).** SR measures whether a policy successfully completes a rollout in the test environment. A rollout is considered successful if the agent does not fall and is able to follow the reference motion with reasonable accuracy. Merely maintaining balance without tracking the reference trajectory is treated as a failure, since such behavior deviates substantially from the intended motion.

**Mean Per Joint Position Error (MPJPE).** MPJPE quantifies the average discrepancy between the predicted and target positions of all joints:

$$\text{MPJPE} = \frac{1}{N} \sum_{i=1}^{N} \left\| \hat{\mathbf{J}}_i - \mathbf{J}_i \right\|_2, \tag{1}$$

where $N$ is the number of joints, $\hat{\mathbf{J}}_i \in \mathbb{R}^3$ denotes the target position of the $i$-th joint, and $\mathbf{J}_i \in \mathbb{R}^3$ is the corresponding predicted position. The unit of measurement is radians.

**Mean Per Keypoint Position Error (MPKPE).** MPKPE evaluates the average error between predicted and target positions of body keypoints:

$$\text{MPKPE} = \frac{1}{K} \sum_{i=1}^{K} \left\| \hat{\mathbf{K}}_i - \mathbf{K}_i \right\|_2, \tag{2}$$

where $K$ is the number of keypoints, $\hat{\mathbf{K}}_i \in \mathbb{R}^3$ represents the target position of the $i$-th keypoint, and $\mathbf{K}_i \in \mathbb{R}^3$ is the predicted position. The unit of measurement is millimeters.

## B Training Details

### B.1 Reward Design

We have listed the rewards used for training the WBC policy and the Delta Action model separately in Table 7 and Table 8, respectively. It is worth noting that when training the Delta Action model, compared to the rewards in ASAP, we used the translation of the root position rather than the positions of all body joints. This is because we did not use a motion capture system, but instead relied on odometry.

Table 8: Reward design for delta action model.

| Term | Expression | Weight |
|------|------------|--------|
| **Penalty** | | |
| Torque limits | $\mathbf{1}(\tau_t \notin [\tau_{\min}, \tau_{\max}])$ | $-10$ |
| DoF position limits | $\mathbf{1}(d_t \notin [q_{\min}, q_{\max}])$ | $-10$ |
| DoF velocity limits | $\mathbf{1}(\dot{d}_t \notin [\dot{q}_{\min}, \dot{q}_{\max}])$ | $-10$ |
| Termination | $\mathbf{1}(\text{termination})$ | $-200.0$ |
| **Regularization** | | |
| DoF acceleration | $\|\ddot{d}_t\|_2^2$ | $-3 \times 10^{-8}$ |
| Action rate | $\|a_t - a_{t-1}\|_2^2$ | $-2$ |
| Action norm | $\|\dot{a}_t\|_2$ | $-2$ |
| Torque | $\|\tau_t\|$ | $-0.0001$ |
| **Task reward** | | |
| Root position | $\exp\left(-4 \cdot \|\hat{p}_{\text{root}} - p_{\text{ref}}\|^2\right)$ | $1.0$ |
| Root rotation | $\exp\left(-4 \cdot \|q_{\text{root}} - q_{\text{ref}}\|^2\right)$ | $0.5$ |
| Root angular velocity | $\exp\left(-4 \cdot \|\omega_{\text{root}} - \omega_{\text{ref}}\|^2\right)$ | $0.5$ |
| Root velocity | $\exp\left(-4 \cdot \|v_{\text{root}} - v_{\text{ref}}\|^2\right)$ | $0.5$ |
| DoF position | $\exp\left(-4 \cdot \|d - d_{\text{ref}}\|^2\right)$ | $0.5$ |
| DoF velocity | $\exp\left(-4 \cdot \|\dot{d} - \dot{d}_{\text{ref}}\|^2\right)$ | $0.5$ |

## B.2 Domain Randomization

Detailed domain randomization setups are summarized in Table 9.

Table 9: The detailed domain randomization implementation. Our types of domain randomization include mechanical properties, external disturbances, and terrain.

| Term | Value |
|---|---|
| **Dynamics Randomization** | |
| Friction | $\mathcal{U}(0.5, 1.25)$ |
| Base CoM offset | $\mathcal{U}(-0.1, 0.1)$ m |
| Link mass | $\mathcal{U}(0.8, 1.2)\times$ default kg |
| P Gain | $\mathcal{U}(0.75, 1.25)\times$ default |
| D Gain | $\mathcal{U}(0.75, 1.25)\times$ default |
| Control delay | $\mathcal{U}(20, 40)$ ms |
| **External Perturbation** | |
| Push robot | interval = 10s, $v_{xy} = 0.5$m/s |
| **Randomized Terrain** | |
| Terrain type | flat, rough |

## B.3 RL Hyperparameters

The RL training progress is aligned with standard PPO [Schulman et al., 2017]. We provide the detailed training hyperparameters in Table 10. We also list the hyperparameters used during the distillation process in Table 11.

Table 10: Hyperparameters for PPO

| Hyperparameter | Value |
|---|---|
| Optimizer | Adam |
| $\beta_1, \beta_2$ | 0.9, 0.999 |
| Learning Rate | $1 \times 10^{-4}$ |
| Batch Size | 4096 |
| Discount factor ($\gamma$) | 0.99 |
| Clip Param | 0.2 |
| Entropy Coef | 0.001 |
| Max Gradient Norm | 1 |
| Learning Epochs | 5 |
| Mini Batches | 4 |
| Value Loss Coef | 1 |

Table 11: Hyperparameters for DAgger

| Hyperparameter | Value |
|---|---|
| Optimizer | Adam |
| Learning Rate | $1 \times 10^{-4}$ |
| Batch Size | 4096 |
| Max Gradient Norm | 1 |
| Learning Epochs | 2 |
| Mini Batches | 4 |

## B.4 Delta Action

For each cluster in each iteration, we randomly sample 20 deployable motions and perform 8 rollouts in the real world. Similar to ASAP, we only train the 4 DoF of the ankles. The average duration per motion is approximately 8 seconds.

## B.5 Training Resource

We used two desktop computers for training. Each was equipped with an Intel i9-13900 CPU, an NVIDIA RTX 4090 GPU, and 64 GB of RAM for policy training.

## C  Model Details

**MLP**  We use a 3-layer MLP model with hidden layer sizes of 1024, 1024, and 512, respectively. The activation function used is ELU.

**Transformer**  Our general WBC controller is built upon the Gated Transformer-XL architecture, adapted from the open-source implementation at https://github.com/datvodinh/ppo-transformer.

The model integrates attention mechanisms with GRU-based gating to enhance memory retention and capture long-range temporal dependencies, making it particularly effective for sequential control tasks in reinforcement learning. The controller takes as input a sequence of 10 consecutive observations and processes them through one Transformer block. Each block employs six attention heads and has a hidden size of 128 and an embedding dimension of 204. The memory length is maintained at 10 to preserve temporal context across sequences.

## D  Additional Results

**Expert Comparison**  As shown in Figure 6, we visualized the comparison between generalists and specialists across six types of clusters. The same trend can still be observed in the remaining two clusters (*Stand Mid* and *Walk Fast*). In both of these two clusters, the policy of the specialists outperforms that of the generalist. However, the final generalist still retains favorable properties and significantly outperforms the initial generalist.

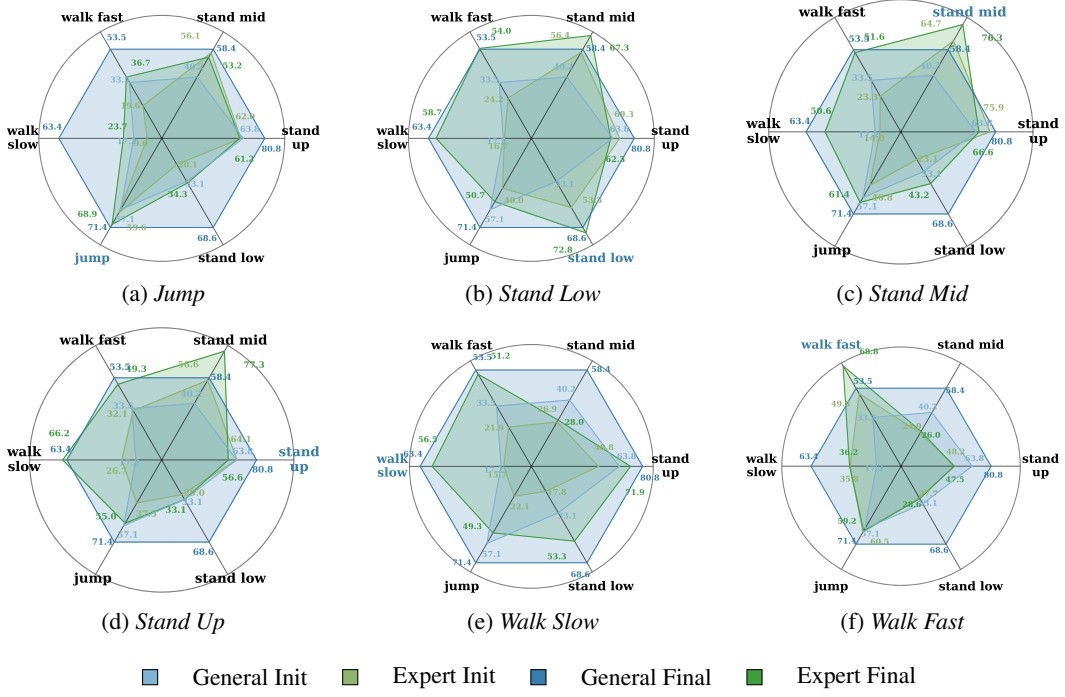

Figure 6: Complete Iterative Comparison.

**Clustering**  We clustered a total of the following number of data samples for each category: *Jump* – 351, *Stand Low* – 229, *Walk Slow* – 3355, *Stand Mid* – 578, *Stand Up* – 2378, and *Walk Fast* – 307. It is evident that the data distribution across the entire AMASS dataset is imbalanced. To balance the distribution for the general strategy, we ensure that each category is equally represented with a ratio of 1/6 during the distillation process.

**Statistical Tests**  To better support our experimental results, we have supplemented Table 12 with the complete confidence intervals (CI) test of the different methods.

Table 12: Main results with confidence interval (CI) statistics reported over samples on a single reference trajectory.

| Method | IsaacGym | | | MuJoCo | | |
|---|---|---|---|---|---|---|
| | SR↑ | MPKPE↓ | MPJPE↓ | SR↑ | MPKPE↓ | MPJPE↓ |
| OmniH2O [He et al., 2024a] | 85.65% (1.114%) | 87.83 (0.6389) | 0.2630 (0.0009) | 15.64% (1.408%) | 360.96 (6.619) | 0.4601 (0.0010) |
| Exbody2 [Ji et al., 2024] | 86.63% (1.106%) | 86.66 (0.6013) | 0.2937 (0.0010) | 50.19% (1.517%) | 272.42 (7.029) | 0.3576 (0.0011) |
| Hover [He et al., 2024c] | 63.21% (1.451%) | 105.84 (0.9350) | 0.2792 (0.0009) | 16.12% (1.491%) | 323.08 (6.919) | 0.3428 (0.0010) |
| **BumbleBee (BB)** | 89.58% (0.946%) | 83.30 (1.1414) | 0.1907 (0.0008) | 66.84% (1.262%) | 294.27 (7.923) | 0.2356 (0.0011) |

