# OpenReview forum: "From Experts to a Generalist: Toward General Whole-Body Control for Humanoid Robots"
_NeurIPS.cc/2025/Conference — NeurIPS 2025 spotlight_

### Official Review · Reviewer_qeFQ · 2025-06-22

**Clarity:** 3
**Significance:** 3
**Originality:** 2
**Rating:** 4
**Confidence:** 4

**Summary:**

This paper proposes BUMBLEBEE (BB), an expert-generalist learning framework that combines motion clusters, trying to overcome challenges that existing frameworks struggle to generalize across highly varied behaviors. Based on the ASAP method, they categorized actions into six major classes and trained an expert policy for each class. Finally, they distilled the six policies into a single general policy using Dagger. The simulation results in the paper outperformed several existing baselines while also successfully completing sim2real experiments.

**Questions:**

1. The lower body in the merged motion demo appears somewhat unstable. What could be causing this behavior?
2. Since the method is modified from the prior method ASAP, it should explicitly use ASAP as the experimental baseline for comparative evaluation. Although the paper compares some algorithms for migrating from H1 to G1 robots, I think a direct comparison with methods primarily designed for G1 robots would be more appropriate.
3. While the proposed method categorizes motions into six classes, how should the system accommodate new action types (e.g., squat) – through additional class expansion or alternative architectural modifications?
4. I observe that ASAP has a more natural jumping effect in a real-world experiment. Are there some examples of comparisons that BB can accomplish but ASAP can't? I think it's important to evaluate the contribution of the BB method.

**Ethical Concerns:**

["NO or VERY MINOR ethics concerns only"]

**Final Justification:**

recommended score: 3->4
The authors have provided clear responses to the questions I raised, which helped resolve my confusion, so I think an additional point should be awarded.

**Limitations:**

yes

**Quality:**

3

**Strengths And Weaknesses:**

Strengths:
1. The proposed method demonstrates successful sim2real deployment for all six categorized action classes, achieving a merged motion combining Charleston Dance and Boxing demo.
2. The paper presents clear explanations, and the methodological logic is fundamentally sound.

Weaknesses:
1. The merged motion demo appears to exhibit lower-body instability in real-world deployment. Without the support of suspension ropes, it remains uncertain whether the system can maintain stable standing.
2. The method in the paper improves upon previous approaches, but it lacks comparison and discussion with prior methods.
3. The experiment does not account for the impact of random seeds on the experimental results.

---

> ### Author Rebuttal · Authors · 2025-07-31
>
> We thank the reviewer for their valuable suggestions. In our response, we primarily address potential misunderstandings, as well as the weaknesses and questions raised in the review.
>
> > Since the method is modified from the prior method ASAP, it should explicitly use ASAP as the experimental baseline for comparative evaluation. Although the paper compares some algorithms for migrating from H1 to G1 robots, I think a direct comparison with methods primarily designed for G1 robots would be more appropriate.
>
> We first address the fundamental differences between our method and the ASAP approach. ASAP is inherently limited to single-motion settings, as it represents task information solely through the motion phase in the observations. This makes direct comparison with our method impossible.
>
> However, to validate the distinction between our method and ASAP, we conduct experiments in which the single delta action model, trained on the entire real dataset, is used to finetune the initial general policy, as described in the paragraph starting at L. 302. As discussed at L. 309, the result shows that this single delta model underperforms in diverse motion settings, supporting our claim that ASAP-style delta modeling does not generalize well across heterogeneous motion classes.
>
> >1. The merged motion demo appears to exhibit lower-body instability in real-world deployment. Without the support of suspension ropes, it remains uncertain whether the system can maintain stable standing.
> >2. The lower body in the merged motion demo appears somewhat unstable. What could be causing this behavior?
>
> The relatively intense lower-body movements observed in the merged motion are primarily due to the task’s high difficulty and large motion range. To maintain whole-body balance, the robot naturally exhibits more compensatory behaviors, especially in complex motion sequences involving boxing. In contrast, for simpler categories such as stand-up, the policy produces highly stable motions, as demonstrated in the two accompanying videos.
>
> Furthermore, across all videos, the rope attached to the robot remains mostly slack, with the rear rope consistently loose. Even when the cabinet moves and the rope trails behind, the resulting force is predominantly horizontal and does not contribute any meaningful support to the robot’s standing behavior. Given that only a single robot was available for the demonstration, we have to employ a rope as a safety measure to protect the hardware.
>
> > While the proposed method categorizes motions into six classes, how should the system accommodate new action types (e.g., squat) – through additional class expansion or alternative architectural modifications?
>
> AMASS already covers a highly diverse range of human motions, and our current six-category division captures a broad spectrum of kinematic behaviors. Importantly, our clustering is based on kinematic characteristics rather than only semantic labels, meaning that any new motion can, in principle, be assigned to one of the existing clusters according to its movement profile—for example, squatting naturally falls under the “stand-low” category in our setting.
>
> Moreover, our expert-to-generalist framework is inherently modular and extensible. The MoE-style structure allows new motion types to be incorporated simply by training additional expert policies tailored to additional category, which can then be distilled into the unified generalist policy. Since only the final distilled model is used for deployment, expert architectures can even differ in network architecture without affecting integration.
>
> > I observe that ASAP has a more natural jumping effect in a real-world experiment. Are there some examples of comparisons that BB can accomplish but ASAP can't? I think it's important to evaluate the contribution of the BB method.
>
> As noted in our previous response regarding the use of ASAP as a baseline, the task settings of ASAP and our method are fundamentally different. BumbleBee is explicitly designed for generalizing across a diverse set of motions, whereas ASAP is tailored for single-motion reinforcement learning. Its policy architecture does not support cross-task generalization, which would require training a separate model for each of the 10,000+ motion sequences in AMASS—an approach that is both computationally prohibitive and impractical.
>
> In contrast, our method is inherently suited for multi-task learning. By clustering motion types based on kinematic characteristics and training expert policies accordingly, BumbleBee can generalize to a wide range of tasks. Moreover, since BumbleBee is trained on over 8,000 motion clips derived from real human data, it demonstrates robust generalization capabilities even without explicit supervision for each individual motion.
>
> > The experiment does not account for the impact of random seeds on the experimental results.
>
> We observe that the variance of policies trained in IsaacGym is very small, consistent with prior works like ASAP and HOVER. This is likely due to the large batch size (envs=4096) used during training, which minimizes randomness across different seeds, resulting in more stable performance. Importantly, the small variance does not cause any overlap and does not affect the results. To keep the main text concise, we have not included full statistical tables, but the complete statistical results are provided in the table below.
>
> |         |     SR     |    MPKPE    |    MPJPE    |     SR     |   MPKPE   |    MPJPE    |
> | ------- | :--------: | :---------: | :---------: | :--------: | :-------: | :---------: |
> | OmniH2O | $\pm$0.14% |  $\pm$0.10  | $\pm$0.0007 | $\pm$0.16% | $\pm$0.80 | $\pm$0.0016 |
> | ExBody2 | $\pm$0.10% |  $\pm$0.15  | $\pm$0.0001 | $\pm$0.41% | $\pm$0.42 | $\pm$0.0025 |
> | Hover   | $\pm$0.27% |  $\pm$0.10  | $\pm$0.0060 | $\pm$0.71% | $\pm$1.11 | $\pm$0.0089 |
> | BB      | $\pm$0.16% | $\pm$0.2252 | $\pm$0.0005 | $\pm$0.17% | $\pm$0.58 | $\pm$0.0011 |
>
> And we will add it into the final version.

---

> > ### Comment · Reviewer_qeFQ · 2025-08-02
> >
> > Thank you for addressing my previous questions. However, I still have a few additional concerns:
> >
> > 1. I understand that in terms of generalization, the BB method indeed addresses the limitations of ASAP as stated in the paper. However, when it comes to individual motion sequences, does BB also achieve better imitation accuracy compared to ASAP, or are their performances nearly equivalent in such cases?
> >
> > 2. I would also like to know the total training time required for the entire teacher-student paradigm based on the AMASS dataset.
> >
> > 3. The paper clusters the motion features of the entire dataset into six categories. Would increasing the number of categories to eight or even ten further benefit the BB method? I’m not entirely sure whether the current clustering approach and category count are sufficient to capture the full range of general motor skills, such as playing ball or dancing.

---

> > > ### Author Response · Authors · 2025-08-04
> > >
> > > We again thank the reviewer for their valuable suggestions.
> > >
> > > >Comparion with ASAP
> > >
> > > We believe the proper way to evaluate BB is under the setting for which it was designed: generalization across diverse motions, which is comprehensively demonstrated in our experiments.
> > >
> > > We respectfully believe that comparing BB with ASAP on individual motion sequences may risk misleading the evaluation of the methodological contributions.
> > >
> > > That said, we acknowledge the reviewer’s curiosity and appreciate the opportunity to clarify: for a single motion sequence, the core components of our method—such as clustering, mixture-of-experts, and expert-to-generalist distillation—are no longer applicable or necessary. In this reduced setting, our pipeline effectively degenerates into a vanilla RL training process, similar in spirit and implementation to ASAP.
> > >
> > > Thank you for raising this point, as it helps us better articulate the scope and design rationale of our method.
> > >
> > > >Training time
> > >
> > > BumbleBee requires 72 hours to converge during training. For comparison, Hover, OmniH2O, and Exbody2 each take approximately 40 hours to reach convergence, while KungfuBot[1] needs roughly 27 hours to train a single motion-tracking policy.
> > >
> > > [1] Xie W, Han J, Zheng J, et al. KungfuBot: Physics-Based Humanoid Whole-Body Control for Learning Highly-Dynamic Skills[J]. arXiv preprint arXiv:2506.12851, 2025.
> > >
> > > >Cluster issues
> > >
> > >  As illustrated in Fig. 3, we selected a moderately balanced number of clusters that offers a practical trade-off between task granularity and implementation complexity. Specifically, beyond this chosen number, the reduction in motion differences within clusters tends to slow down as the number of clusters increases, meaning adding more clusters yields diminishing returns in terms of distinguishing finer motion nuances.
> > >
> > > In essence, clustering is used to simplify the learning problem: with more clusters, each expert is trained on a narrower and easier subtask, which typically improves expert performance. However, increasing the number of clusters also significantly raises the overall workload and complicates the distillation process. Given this trade-off, we opted for a reasonable balance in our current setup.
> > >
> > > The experimental results show that this cluster count yields better performance compared to training on the entire dataset. It effectively enables meaningful distinctions: for instance, motions like "playing ball" and "walk" are generally grouped under "Walk-fast or Walk-slow" and at the very least, separating such categories from more still motions simplifies the overall task significantly.

---

> > > > ### Author Response · Authors · 2025-08-05
> > > >
> > > > Thank you for your response. If you have any further questions or suggestions, please feel free to let us know and we are happy to continue the discussion and provide additional clarifications as needed.

---

> > > > > ### Comment · Reviewer_qeFQ · 2025-08-05
> > > > >
> > > > > Thank you for your response. I think your answer has largely addressed the confusion I had before, so I will raise my score as an acknowledgment of this work.

---

> > > > > > ### Author Response · Authors · 2025-08-08
> > > > > >
> > > > > > Thank you for your valuable comments and detailed feedback. We will incorporate the content of our rebuttal into the main text to address the points raised, ensuring greater clarity and completeness in the revised version.

---

### Official Review · Reviewer_a9GS · 2025-06-27

**Clarity:** 4
**Significance:** 3
**Originality:** 3
**Rating:** 5
**Confidence:** 4

**Summary:**

The paper describes an approach for the learning of classes of movements for a humanoid robot which is based on a combination of a clustering of the training movements using an autoencoder approach, and the learning of controller models for the movement subclasses, whose outputs are combined using a mixtures of expert approach. The subclass movement policies are learned by reinforcement learning, where robustness and sim-to-real transfer is ensured by an action delta learning approach. The movements are implemented on a real Unitree G1 robot. It is demonstrated convincingly that the new approach outperforms other SOTA methods for different types of walking, standing and jumping movements. Also the efficiency of the movement class-specific delta action learning is demonstrated in comparison with approaches that learn only a single motion policy model.

**Questions:**

Does it always make sense to take semantic similarity into account? It appears that it might be somewhat dependent on the quality of the semantic labeling and the specificity of semantic labeling. For example, 'turning' might include a whole range of movements that are much more variable than different types of walking (slow fast etc.) Can it be guessed how the method would, extend for such more heterogeneous motion classes ?


Minor:
- What is the meaning of k(s) in L. 191. Is k an index?

- L. 319: There is the class 'RUN' mentioned that seems not to appear anywhere before. Is this fast walking?

**Ethical Concerns:**

["NO or VERY MINOR ethics concerns only"]

**Final Justification:**

My concerns were adequately answered, and so were most remarks of the other reviewers. I thus maintain my score (accept).

**Limitations:**

yes in principle; one could say a few words about scaling to larger and more heterogeneous data sets, where I fully understand that implementing all this on real robots wold be a lot of work.

**Quality:**

4

**Strengths And Weaknesses:**

The paper is very well written and organized. The chosen approach is straight forward and makes sense. The superiority of the methods is convincingly shown, and there is a careful analysis also of the reasons why the mixture of experts method is superior to other approaches.
Also the presented comparison with other baseline methods using multiple different simulators is convincing.

A thing that might be interesting to explore i the future is how the quality of the method changes with the number of clusters, and
maybe even more interesting, how the methods scales for more classes of substantially different movements.

---

> ### Author Rebuttal · Authors · 2025-07-31
>
> We appreciate the reviewer’s feedback and have focused our response on addressing the aspects of the paper that may require improvement.
>
> > A thing that might be interesting to explore i the future is how the quality of the method changes with the number of clusters, and maybe even more interesting, how the methods scales for more classes of substantially different movements.
> >
> > Does it always make sense to take semantic similarity into account? It appears that it might be somewhat dependent on the quality of the semantic labeling and the specificity of semantic labeling. For example, 'turning' might include a whole range of movements that are much more variable than different types of walking (slow fast etc.) Can it be guessed how the method would, extend for such more heterogeneous motion classes ?
>
> As illustrated in Fig. 3, we selected a moderately balanced number of clusters that offers a practical trade-off between task granularity and implementation complexity. In essence, clustering is used to simplify the learning problem: with more clusters, each expert is trained on a narrower and easier subtask, which typically improves expert performance. However, increasing the number of clusters also significantly raises the overall workload and complicates the distillation process. Given this trade-off, we opted for a reasonable balance in our current setup. We acknowledge the importance of this parameter and will make an effort to investigate its impact in the final version.
>
> We share the same concern regarding semantic similarity, as language-based labels can suffer from quality issues, ambiguity, and redundancy. For instance, the label "playing basketball" may correspond to a range of motions, such as dribbling while moving or standing still to shoot, which are kinematically distinct despite sharing the same semantic tag. To address this, we do not rely directly on semantic labels for clustering. Instead, we align motion and text into a shared latent space and perform clustering based on this representation. This allows for a more meaningful categorization that captures both fine-grained motion differences and underlying semantic relationships, as illustrated in Table 2.
>
> > - What is the meaning of k(s) in L. 191. Is k an index?
>
> Here, *k* refers to the index of the expert corresponding to the motion that the given state belongs to. Thank you for pointing out and we will revise it accordingly.
>
> > - L. 319: There is the class 'RUN' mentioned that seems not to appear anywhere before. Is this fast walking?
>
> We appreciate your observation. In this case, the label "run" was indeed intended to represent a fast walking motion. We will revise it in the final version.

---

> > ### Comment · Reviewer_a9GS · 2025-08-05
> > **Thanks for changes**
> >
> > I thank the authors for carefully considering my comments. Their answers seems meaningful to me, and assuming that my smaller requests for corrections are implemented in the final manuscript, I have no further comments or requests.

---

> > > ### Author Response · Authors · 2025-08-08
> > >
> > > Thank you for your valuable comments and detailed feedback. We will incorporate the content of our rebuttal into the main text to address the points raised, ensuring greater clarity and completeness in the revised version.

---

### Official Review · Reviewer_sTVB · 2025-07-02

**Clarity:** 3
**Significance:** 3
**Originality:** 3
**Rating:** 5
**Confidence:** 4

**Summary:**

This paper studies the problem of learning generalist whole-body control policies for humanoid robots. Recent work has demonstrated that whole-body control policies trained to imitate reference motions (re-targeted to a robot embodiment) in simulation can be transferred to real humanoid robots, either as-is (zero-shot sim2real) or with a small amount of finetuning/adaptation to real world data (residual action/dynamics modeling). While several works have considered generalist (multi-clip) whole-body control / tracking for simulated human embodiments (e.g. MoCapAct), there is comparatively little work on transferring such generalist policies to real humanoid robots. This work considers exactly that problem, and proposes an expert-to-generalist learning framework -- dubbed Bumblebee -- that successfully learns multi-clip whole-body control policies that transfer to the real world. The general idea is to (1) cluster all reference motions by semantic and behavioral similarity using an autoencoder, and (2) train a generalist tracking policy on the full set of motions which can then (3) be finetuned on each motion cluster. Then, (4) each cluster expert is deployed in the real world and (5) a per-cluster delta control policy is obtained with real data. Finally, (6) the cluster expert and sim2real delta policies are then distilled into a single Transformer-based generalist policy via DAgger. Experiments indicate that Bumblebee successfully learns a multi-clip generalist policy that can perform a variety of motions on a real humanoid robot, while also performing comparably or better in simulated motion tracking tasks than recent related works.

**Questions:**

I would like the authors to please address my comments listed under "weaknesses" in the previous section. I believe that they are all rather concrete and actionable.

**Ethical Concerns:**

["NO or VERY MINOR ethics concerns only"]

**Final Justification:**

The majority of my concerns have been addressed by the authors' rebuttal. I am willing to raise my score and recommend acceptance on the condition that the authors fully incorporate all reviewer feedback into their camera-ready version.

**Score change: 4->5**

**Confidence change: 3->4**

**Limitations:**

As per my previous comment:
> The discussion of limitations is rather minimal and merely points out that additional sensors are likely to improve results. I strongly suggest that the authors revise their limitation section to discuss a few key limitations pertaining to the proposed method itself, e.g. the way that clustering is currently performed, or the scalability of the generalist-to-expert-to-generalist training pipeline.

**Quality:**

2

**Strengths And Weaknesses:**

This paper was a pleasant read overall. While my preliminary assessment leans positive, I identify several aspects of the paper that can be improved. I summarize my thoughts as follows:

**Strengths**
- The paper is generally well organized and easy to follow. The problem of generalist whole-body humanoid control on real hardware is an interesting and timely problem, and the paper is thus likely to be of interest to researchers at the intersection of machine learning and robotics. The paper is self-contained and contains sufficient discussion of recent related work. Figures and illustrations contribute positively to my understanding of the approach and results.
- The proposed method, while rather complex due to it consisting of multiple stages, is composed of a number of intuitive techniques that are well established in the literature (clustering, generalist-to-expert finetuning, residual policies, DAgger, etc.). The overall approach is thus fairly intuitive to readers familiar with the area; I consider this a strength. I appreciate that the method is described in great detail.
- The experimental setup appears to be fairly rigorous and includes several standard (quantitative) performance metrics in simulation (success rate, joint/coordinate tracking errors). Although the real-world evaluation is rather limited, I do appreciate the inclusion of qualitative results on real hardware.

**Weaknesses**
- As alluded to above, it would be rather insightful if the authors could provide some quantitative evaluation metrics for the hardware experiments. I understand that computing the same performance metrics as in simulation may be difficult, but on the other hand it is also difficult to evaluate the effectiveness of the expert refinement + final generalist performance without any metrics or baselines / comparisons.
- I am somewhat surprised to see such a large gap in performance between methods in MuJoCo when numbers are rather close in IsaacGym. The authors argue that this is because MuJoCo is a more realistic simulator, but I find that to be a rather handwavy explanation for such a dramatic difference in results. Could the authors please elaborate on why there might be such a discrepancy between the two on a more technical level, and e.g. include some videos of each method evaluated in MuJoCo? Do baselines tend to overfit to the IsaacGym simulator? If so, could that possibly be alleviated by improving policy robustness with domain randomization or added control noise?
- Given that clustering places a key role in this work, I expected to see an ablation on the number of clusters. I understand that this could potentially be quite time consuming if it includes real world policy refinement, so perhaps a simpler ablation could be set up purely in simulation. Even so, this might not be feasible within the strict time frame of a rebuttal. It would be helpful to include some discussion on the (expected) effect of the number of clusters on the overall approach, ideally backed by empirical evidence for e.g. the camera-ready version.
- Unfortunately, I have not been able to view any of the supplemental videos. I tried two different browsers (Chrome, Firefox) and have also tried downloading the repo but it keeps failing. I give the authors the benefit of the doubt here but it is rather unfortunate nonetheless.
- I like the general idea behind Figure 4; it neatly summarizes the empirical expert vs. generalist results. However, the different methods are currently difficult to visually distinguish due to the choice of colors. I would recommend refining this figure to better separate the different methods visually.
- The discussion of limitations is rather minimal and merely points out that additional sensors are likely to improve results. I strongly suggest that the authors revise their limitation section to discuss a few key limitations pertaining to the proposed method itself, e.g. the way that clustering is currently performed, or the scalability of the generalist-to-expert-to-generalist training pipeline.

---

> ### Author Rebuttal · Authors · 2025-07-31
>
> ## To reviewer sTVB
>
> Thanks for your detailed reviews.
>
> ### Hardware experiments
>
> > - As alluded to above, it would be rather insightful if the authors could provide some quantitative evaluation metrics for the hardware experiments. I understand that computing the same performance metrics as in simulation may be difficult, but on the other hand it is also difficult to evaluate the effectiveness of the expert refinement + final generalist performance without any metrics or baselines / comparisons.
>
> We agree with that it is important to provide quantitative evaluation metrics for the hardware experiments.
>
> Also, it is challenging to provide precise quantitative comparisons for real-world deployment, primarily due to the lack of publicly available deployment code for baseline methods, which makes fair sim-to-real comparisons difficult. To address this, we present extensive videos demonstrating the capabilities of our final policy and comparisons with and without delta fine-tuning in each cluster. Additionally, we report results for our final general policy on 100 randomly sampled motions from the AMASS dataset, achieving a success count of 59 and an MPJPE of 0.2896 — slightly lower than its performance in MuJoCo, but still demonstrating strong real-world generalization.
>
>    ## Gap between IsaacGym and MuJoCo
>
> > - I am somewhat surprised to see such a large gap in performance between methods in MuJoCo when numbers are rather close in IsaacGym. The authors argue that this is because MuJoCo is a more realistic simulator, but I find that to be a rather handwavy explanation for such a dramatic difference in results. Could the authors please elaborate on why there might be such a discrepancy between the two on a more technical level, and e.g. include some videos of each method evaluated in MuJoCo? Do baselines tend to overfit to the IsaacGym simulator? If so, could that possibly be alleviated by improving policy robustness with domain randomization or added control noise?
>
> We appreciate the reviewer’s thoughtful question regarding the performance discrepancy between MuJoCo and IsaacGym. We believe several factors contribute to this difference.
>
> 1. **The physics engine**.  The discrepancy in results between MuJoCo and IsaacGym can be partially attributed to differences in their physical simulation fidelity. Specifically, MuJoCo provides more accurate and tunable contact modeling, which are often approximated more coarsely in IsaacGym. Additionally,  options such as `collapse_fixed_joints=True` are often used to reduce model complexity and improve simulation speed. While this can be beneficial for large-scale training, it effectively removes certain physical structures that are present in the original URDF models. Furthermore, joint-level control in IsaacGym may suffer from limited temporal precision, making it challenging to accurately regulate the duration and timing of torque application. In contrast, MuJoCo provides fine-grained control over both the timing and magnitude of applied torques, which is essential for precise motor behaviors. Finally, MuJoCo offers detailed control over joint friction and damping. All above can significantly impact performance in tasks especially **involving locomotion or contact-rich behavior** that account for a significant portion of the AMASS dataset.
>
> 2. **Observation space.**   We hypothesize that different observation space designs exhibit varying levels of robustness in domain transfer scenarios. Compared to the local-coordinate-based observation space used in our method and ExBody2, world-frame-based observations—such as absolute keypoints positions—are significantly more fragile. This is because even minor perturbations or mismatches in the environment (e.g., initial pose differences) are directly reflected in the observations, amplifying their impact on policy performance. In our experiments, MuJoCo initializes the robot to a default pose that approximates real-world joint configurations, while IsaacGym resets the robot based on the reference motion. This discrepancy introduces an initial state mismatch between the two simulators. As a result, policies relying on world-frame position, such as absolute keypoints positions, suffer greater performance degradation under domain transfer, as also observed in ExBody2 for contrast. This issue becomes even more critical in real-world deployment, where precise resetting to a reference pose is generally infeasible.
>
> 3. **Overfitting in IsaacGym**.  We acknowledge that overfitting during training in IsaacGym is a potential concern. In our experiments, we applied observation noise and domain randomization to the same extent as in ASAP. However, excessively large noise or overly aggressive domain randomization can significantly degrade training performance, and identifying the optimal trade-off remains a well-known challenge across all robot control algorithms. ASAP itself has been widely recognized as an effective approach for sim-to-real transfer. Building on this foundation, we extend the framework by introducing motion category diversity, enabling the learned general tracking policy to achieve better domain transferability.
>
> We apologize for not being able to include additional figures at the rebuttal stage. However, as shown in Fig. 5, similar issues are observed in the MuJoCo environment as well — the policy often exhibits noticeable foot instability, struggles to accurately follow the reference motions, and in some cases, crashes early due to significant initial state mismatch. These issues may be attributed to the factors discussed above.
>
> ### Number of clusters
>
> >- Given that clustering places a key role in this work, I expected to see an ablation on the number of clusters. I understand that this could potentially be quite time consuming if it includes real world policy refinement, so perhaps a simpler ablation could be set up purely in simulation. Even so, this might not be feasible within the strict time frame of a rebuttal. It would be helpful to include some discussion on the (expected) effect of the number of clusters on the overall approach, ideally backed by empirical evidence for e.g. the camera-ready version.
>
> Thank you for the suggestion. We agree that exploring different numbers of clusters is valuable, but it also involves substantial additional effort. As illustrated in Fig. 3, we selected a moderately balanced number of clusters that offers a practical trade-off between task granularity and implementation complexity. In essence, clustering is used to simplify the learning problem: with more clusters, each expert is trained on a narrower and easier subtask, which typically improves expert performance. However, increasing the number of clusters also significantly raises the overall workload and complicates the distillation process. Given this trade-off, we opted for a reasonable balance in our current setup. That said, we acknowledge the importance of this parameter and will make an effort to investigate its impact in the final version.
>
> ### Supplemental videos
>
> We apologize for the inconvenience. You may try accessing the alternative link provided in the anonymous link, or download the video directly. This is currently the best method we can offer to share the video anonymously.
>
> ### Suggestions for the paper
>
> > I like the general idea behind Figure 4; it neatly summarizes the empirical expert vs. generalist results. However, the different methods are currently difficult to visually distinguish due to the choice of colors. I would recommend refining this figure to better separate the different methods visually.
>
> 1. Thank you for pointing this out. The data presented in Fig. 4 is indeed information-dense, and the color scheme may be too similar to distinguish clearly. We will revise the figure in the final version to improve clarity.
>
> > The discussion of limitations is rather minimal and merely points out that additional sensors are likely to improve results. I strongly suggest that the authors revise their limitation section to discuss a few key limitations pertaining to the proposed method itself, e.g. the way that clustering is currently performed, or the scalability of the generalist-to-expert-to-generalist training pipeline.
>
> 2. We appreciate your valuable feedback. In the revised version, we add some of the method's limitations  pertaining to the proposed method itself, including the two issues you mentioned, the overall methodological complexity, and the significant amount of real-world iteration that may be required for practical deployment.

---

> > ### Comment · Reviewer_sTVB · 2025-08-04
> > **Thank you**
> >
> > Thank you for the detailed answers to concerns raised in my initial review.
> >
> > > To address this, we present extensive videos demonstrating the capabilities of our final policy and comparisons with and without delta fine-tuning in each cluster. Additionally, we report results for our final general policy on 100 randomly sampled motions from the AMASS dataset, achieving a success count of 59 and an MPJPE of 0.2896 — slightly lower than its performance in MuJoCo, but still demonstrating strong real-world generalization.
> >
> > Thanks for providing these numbers. I understand that some degradation is to be expected when transferring to real; the concern that I had was moreso the fact that no such quantitative metrics were reported at all. Including just a few quantitative evaluations in the real world in the camera-ready version should be quite helpful for readers to gauge the expected transfer performance beyond a few videos that may not be representative.
> >
> > Regarding poor baseline performance in MuJoCo:
> >
> > > The physics engine
> >
> > It is to be expected that this would influence results to some extent, and is the main argument made in the paper.
> >
> > > Observation space
> >
> > This definitely seems to be a key reason for the poor performance. It would be really helpful for readers if this would be made clear in the paper.
> >
> > > Overfitting in IsaacGym
> >
> > Again, this is expected to some extent, but it would be useful to make that same written argument in the paper (you already randomize to some degree but that may not be sufficient to transfer to a new simulator without hampering performance in any way).
> >
> > > Number of clusters
> >
> > Thanks for acknowledging that this would be informative. I understand that a comprehensive ablation would be rather demanding in terms of resources, but trying at least 1 other number of clusters and including that comparison in the camera-ready should certainly be feasible. Given that clustering is quite central to the approach, I think that is necessary.
> >
> > > In the revised version, we add some of the method's limitations pertaining to the proposed method itself
> >
> > Thank you for being committed to revising the limitations section. All methods have limitations, and being transparent about what those limitations are is extremely valuable to readers and anyone looking to build upon your work.
> >
> > **In summary, the majority of my concerns have been addressed by the authors' rebuttal. I am willing to raise my score and recommend acceptance on the condition that the authors fully incorporate all reviewer feedback into their camera-ready version.**

---

> > > ### Author Response · Authors · 2025-08-05
> > >
> > > Thank you for your valuable comments and detailed feedback. We will incorporate the content of our rebuttal into the main text to address the points raised, ensuring greater clarity and completeness in the revised version.

---

### Official Review · Reviewer_4Wt3 · 2025-07-02

**Clarity:** 3
**Significance:** 3
**Originality:** 3
**Rating:** 5
**Confidence:** 3

**Summary:**

The paper proposes BUMBLEBEE (BB), a method for learning real-robot motion tracking tasks. The method analyzes human motion capture data, autoencoding motion data and textual descriptions in order to cluster motions into 6 clusters.

BB then trains a general motion tracking policy, then for each motion cluster, it finetunes an “expert” for specifically those motions, starting from the general policy. It also trains action delta policies to reduce the sim-to-real gap, using data from rolling out simulated policies on real robots.

Finally, BB distills the experts into a single policy, which is shown to perform better than prior methods on motion tracking metrics. The final real-robot policy successfully performs highly dynamic actions.

**Questions:**

1) Based on SR as show in Table 3, “General Init” already outperforms all baselines on IsaacGym, and all baselines except Exbody2 on MuJoCo. How much of the benefit of BB comes simply from improved NN architectures/other hyperparameter choices, vs. the clustering, expert, and distillation steps emphasized in the paper? It would be good to include the full set of metrics for “General Init” in Table 1. If most of the benefit is coming from differences other than the clustering approach, these differences should be described in more detail.

2) While the paper includes ablations showing that the use of multiple experts improves performance over the generalist policy (e.g. Table 3), it’s still not clear to me why this is the case. Do the generalist and 6 experts all use the same model architecture described in Appendix C? If so, maybe the greater total capacity of the experts allows better performance, which can then be distilled. What if, instead of 6 separate experts, you trained a single “General Init” using a model with 6x the total parameter count, then distilled this down to a single model of the original size? This would eliminate the need for the clustering models and significantly simplify the full method; unclear based on the paper whether we should expect this to work?

**Ethical Concerns:**

["NO or VERY MINOR ethics concerns only"]

**Final Justification:**

The authors added statistical testing, addressing my concern.

As discussed with the authors, insofar as the IsaacGym results are demonstrating anything, they're primarily demonstrating the effectiveness of the authors' hyperparameter tuning compared to prior work, rather than the performance of the BB algorithm per se. I believe the authors should make this clearer in the writeup. Including General Init in Table 1 as discussed will help here.

**Limitations:**

Yes

**Quality:**

3

**Strengths And Weaknesses:**

**Strengths:**
1) The real-robot videos are very impressive! The robot stays balanced even during very rapid motions.
2) BB is shown to improve motion tracking performance compared to three other recent methods: OmniH2O, Exbody2, and Hover.
3) Detailed analysis of the method: each component is explained and ablated.


**Weaknesses:**
1) The full method is highly complex. While the paper’s ablations show that each component adds at least some performance compared to its absence, the importance of certain design decisions is still unclear; see questions.
2) Results lack CIs and statistical tests for group differences (Tables 1-6).
3) Some unclear environment descriptions: what is “Success Rate”? The paper says “Among all metrics, SR is the most critical, as it reflects the overall viability and stability of the policy” and “Detailed definitions of these metrics can be found in Appendix A.”, but I’m not able to find these definitions in the provided appendix A.

---

> ### Author Rebuttal · Authors · 2025-07-31
>
> ## To reviewer 4Wt3
>
> We sincerely thank Reviewer 4Wt3 for the insightful comments. We mainly clarify the points that may have caused confusion and address the weaknesses.
>
> ### Clarification for design decisions
>
> > How much of the benefit of BB comes simply from improved NN architectures/other hyperparameter choices, vs. the clustering, expert, and distillation steps emphasized in the paper? It would be good to include the full set of metrics for “General Init” in Table 1. If most of the benefit is coming from differences other than the clustering approach, these differences should be described in more detail.
>
> Thank you for the thoughtful question. To ensure fairness, we kept the MLP architecture consistent across all models, including ablations and baselines. The only exception is **BUMBLEBEE**, where we employed a Transformer-based architecture **specifically for the purpose of distilling multiple experts more effectively**. And thank you for your remind, we will include the "General Init" into Table 1. We also conduct an additional experiment that use the same MLP as backbone of BB and get the success rate result of 89.56% in IsaacGym and 61.04% in MuJoCo. The MuJoCo result is slightly lower than BB, but it is still significantly higher than the baseline, which proves that while the architecture does have an impact, the main influence comes from the expert and clustering strategies.
>
>
> ### Statistical tests
>
> We observe that the variance of policies trained in IsaacGym is very small, consistent with prior works like ASAP and HOVER. This is likely due to the large batch size (envs=4096) used during training, which minimizes randomness across different seeds, resulting in more stable performance. Importantly, the small variance does not cause any overlap and does not affect the results. To keep the main text concise, we have not included full statistical tables, but the complete statistical results are provided in the table below.
>
> |         |     SR     |    MPKPE    |    MPJPE    |     SR     |   MPKPE   |    MPJPE    |
> | ------- | :--------: | :---------: | :---------: | :--------: | :-------: | :---------: |
> | OmniH2O | $\pm$0.14% |  $\pm$0.10  | $\pm$0.0007 | $\pm$0.16% | $\pm$0.80 | $\pm$0.0016 |
> | ExBody2 | $\pm$0.10% |  $\pm$0.15  | $\pm$0.0001 | $\pm$0.41% | $\pm$0.42 | $\pm$0.0025 |
> | Hover   | $\pm$0.27% |  $\pm$0.10  | $\pm$0.0060 | $\pm$0.71% | $\pm$1.11 | $\pm$0.0089 |
> | BB      | $\pm$0.16% | $\pm$0.2252 | $\pm$0.0005 | $\pm$0.17% | $\pm$0.58 | $\pm$0.0011 |
>
> And we will add it into the final version.
>
> ### Parameter count
>
> > While the paper includes ablations showing that the use of multiple experts improves performance over the generalist policy (e.g. Table 3), it’s still not clear to me why this is the case. Do the generalist and 6 experts all use the same model architecture described in Appendix C? If so, maybe the greater total capacity of the experts allows better performance, which can then be distilled. What if, instead of 6 separate experts, you trained a single “General Init” using a model with 6x the total parameter count, then distilled this down to a single model of the original size? This would eliminate the need for the clustering models and significantly simplify the full method; unclear based on the paper whether we should expect this to work?
>
> As requested, we conducted an additional experiment by increasing the parameter size by 6× while keeping all other settings identical to the "General init" configuration. The success rates were 85.17% in IsaacGym and 37.36% in MuJoCo, showing minimal performance improvement over the original model. This further confirms that simply increasing the model size has limited impact on performance.
>
> ### Detailed description of the metric
>
> > Some unclear environment descriptions: what is “Success Rate”? The paper says “Among all metrics, SR is the most critical, as it reflects the overall viability and stability of the policy” and “Detailed definitions of these metrics can be found in Appendix A.”, but I’m not able to find these definitions in the provided appendix A.
>
> Thank you for the feedback. In our setting, "Success Rate" (SR), shown in line 813 of appendix, refers to the policy successfully completing the task without failure. More specifically, a trial is considered successful only if the agent maintains balance (i.e., does not fall) and closely follows the reference motion. A policy that remains stable but deviates significantly from the target motion would still be considered a failure. We will include SR and the two additional metrics in Appendix A.
>
> The **Mean Per Joint Position Error (MPJPE)**:
>
> $$
> \text{MPJPE} = \frac{1}{N} \sum_{i=1}^{N} \left\| \hat{\mathbf{J}}_i - \mathbf{J}_i \right\|_2
> $$
>
> - $N$: Number of joints
> - $\hat{\mathbf{J}}_i \in \mathbb{R}$: Target joint position of the $i$-th joint
> - $\mathbf{J}_i \in \mathbb{R}$: Joint position of the $i$-th joint.
> - Unit: rad
>
> The **Mean Per Keypoint Position Error (MPKPE)** :
>
> $$
> \text{MPKPE} = \frac{1}{N} \sum_{i=1}^{N} \left\| \hat{\mathbf{K}}_i - \mathbf{K}_i \right\|_2
> $$
>
> - $N$: Number of keypoints
> - $ \hat{\mathbf{K}}_i \in \mathbb{R}^3 $: Target position of the $i$-th keypoint
> - $\mathbf{K}_i \in \mathbb{R}^3 $: Position of the $i$-th keypoint
> - Unit: mm

---

> > ### Comment · Reviewer_4Wt3 · 2025-08-01
> >
> > Thanks for your responses.
> >
> > **Design decisions**
> >
> > I’m still a bit confused - “General Init” as shown in Table 3 appears to outperform all baselines other than BUMBLEBEE on IsaacGym in Table 1, and all baselines other than Exbody2 on MuJoCo. If the performance improvement of General Init over other methods isn’t coming from the clustering/experts and isn’t coming from an architectural difference, where is it coming from?
> >
> > **Statistical tests**
> >
> > IIUC you’re evaluating performance tracking AMASS trajectories, right? Can you compute CIs with respect to individual AMASS trajectories? I think this is the typical approach for ML evaluation - compute CIs with respect to dataset examples. (Otherwise deterministic policies would have 0-width CIs).
> >
> > If you want to use multiple policy rollouts per reference trajectory, the individual rollouts won’t be IID, but you can use clustered errors as described in https://www.anthropic.com/research/statistical-approach-to-model-evals. Alternatively, you could treat the model's average performance over samples on a single reference trajectory as one IID sample, and use a standard statistical test like a bootstrap test (https://docs.scipy.org/doc/scipy/reference/generated/scipy.stats.bootstrap.html) over examples.
> >
> > **Parameter count**
> >
> > Thanks, this is a useful experiment and falsifies my hypothesis. The paper’s cross-task interference hypothesis seems consistent with the evidence.
> >
> > **Description of the metric**
> >
> > Thanks, adding your given definition of SR and the additional metrics to Appendix A addresses my concern.

---

> > > ### Author Response · Authors · 2025-08-03
> > >
> > > Thank you for your feedback. We will continue to clarify any points that might cause confusion.
> > >
> > > ### **General Init**
> > >
> > > On IsaacGym, our base policy performs comparably to Exbody2 and OmniH2O. Since all three approaches are trained under similar conditions and belong to the same performance tier, the minor differences observed are within reasonable variation and likely attributable to differences in hyperparameter tuning. These small numerical gaps are insufficient to draw definitive conclusions about superiority among them.
> > >
> > > On the other hand, the improved performance of General Init in MuJoCo can be largely attributed to its use of local observation spaces, as opposed to global keypoint tracking used in earlier methods like H2O and OmniH2O:
> > >
> > > Exbody2 points out that previous whole-body tracking methods, such as H2O, OmniH2O, and HOVER, rely on global keypoint positions. This reliance often leads to error accumulation, especially when generalizing to new domains. As the robot struggles to precisely follow global reference poses in unfamiliar environments, even small deviations in tracking can quickly compound over time, resulting in failure to complete the motion. In contrast, Exbody2 transforms keypoints into the local coordinate frame and decouples posture tracking from velocity control, significantly improving robustness and generalization in dynamic settings.
> > >
> > > In line with this observation, our method also adopts a local-coordinate observation design, which proves more robust under sim-to-real domain shifts. Our evaluations in MuJoCo, particularly the performance of General Init, BB, and Exbody2, further support the importance of using local reference frames when deploying policies across diverse and unseen motion scenarios.
> > >
> > >
> > > ### **Statistical tests**
> > >
> > > Apologies for the earlier misunderstanding. To clarify, should we treat the AMASS dataset as the test set and compute confidence intervals (CIs) over different motions, considering each motion as a random variable? If that is the case, our statistical results for 95% CIs are shown in the following table:
> > >
> > > |         | SR              | MPKPE          | MPJPE          | SR             | MPKPE         | MPJPE          |
> > > | ------- | --------------- | -------------- | -------------- | -------------- | ------------- | -------------- |
> > > | OmniH2O | 85.65% (1.114%) | 87.83(0.6389)  | 0.2630(0.0009) | 15.64%(1.408%) | 360.96(6.619) | 0.4601(0.0010) |
> > > | ExBody2 | 86.63%(1.106%)  | 86.66(0.6013)  | 0.2937(0.0010) | 50.19%(1.517%) | 272.42(7.029) | 0.3576(0.0011) |
> > > | Hover   | 63.21%(1.451%)  | 105.84(0.9350) | 0.2792(0.0009) | 16.12%(1.491%) | 323.08(6.919) | 0.3428(0.0010) |
> > > | BB      | 89.58%(0.946%)  | 83.30(1.1414)  | 0.1907(0.0008) | 66.84%(1.262%) | 294.27(7.923) | 0.2356(0.0011) |
> > >
> > > We compute the confidence interval using the formula: $z*\frac{\sigma}{\sqrt{N}}$, where z = 1.96 for a 95% confidence level, N is the number of motions, and $\sigma$ is the standard deviation of the success rates, MPKPE, and MPJPE across motions.

---

> > > > ### Comment · Reviewer_4Wt3 · 2025-08-04
> > > >
> > > > Thanks for your responses.
> > > >
> > > > **General Init**
> > > >
> > > > Understood, thanks for clarifying. On IsaacGym, the gaps between BB and General Init are smaller than the gaps between General Init and all other methods. It seems reasonable to say that "the minor differences observed are within reasonable variation and likely attributable to differences in hyperparameter tuning", but if this is the case, it's important to be clear that insofar as the IsaacGym results are demonstrating anything, they're primarily demonstrating the effectiveness of the authors' hyperparameter tuning compared to prior work, rather than the performance of the BB algorithm per se. I believe the authors should make this clearer in the writeup. Including General Init in Table 1 will help here.
> > > >
> > > > I agree that the MuJoCo results show a much clearer improvement from BB.
> > > >
> > > > **Statistical Tests**
> > > >
> > > > Thanks! This resolves my concern. Please do report these CIs in the camera ready.

---

> > > > > ### Author Response · Authors · 2025-08-05
> > > > >
> > > > > Thank you for your valuable comments and detailed feedback. We will incorporate the content of our rebuttal into the main text to address the points raised, ensuring greater clarity and completeness in the revised version.

---

### Decision · Program_Chairs · 2025-09-17

**Decision:**

Accept (spotlight)

**Comment:**

Reviewers agree that this is a significant step forward in the field. The rebuttal and post-rebuttal discussion clarified initial questions. Congratulations, this submission is accepted.